# Identification and Characterization of the *LecRLKs* Gene Family in Maize, and Its Role Under Biotic and Abiotic Stress

**DOI:** 10.3390/biology14010020

**Published:** 2024-12-28

**Authors:** Xiangbo Yang, Ziqi Chen, Jianyu Lu, Xuancheng Wei, Yanying Yao, Wendi Lv, Jiarui Han, Jianbo Fei

**Affiliations:** 1College of Agriculture, Jilin Agricultural Science and Technology College, Jilin 132101, China; yangxiangbo1980@163.com; 2Institute of Agricultural Biotechnology/Jilin Provincial Key Laboratory of Agricultural Biotechnology, Jilin Academy of Agricultural Sciences (Northeast Innovation Center of Agricultural Science and Technology in China), Changchun 130033, China; chenziqi@cjaas.com (Z.C.); weixuancheng2022@163.com (X.W.); hdrplybest@163.com (J.H.); 3Institute of Agricultural Biotechnology, Jilin Agricultural University, Changchun 130117, China; lu@mails.jlau.edu.cn; 4Jilin Provincial Agricultural Environmental Protection and Rural Energy Management General Station, Changchun 130033, China; jlnhhyyy@163.com (Y.Y.); lvwendi1005@163.com (W.L.)

**Keywords:** maize, LecRLK gene family, phylogeny analysis, subcellular localization, expression analysis

## Abstract

Biological and abiotic stresses are significant factors that threaten maize yield. We employed whole-genome analysis to identify the lectin receptor-like kinase family. Phylogenetic analysis of this gene family, along with evaluations of chromosomal location, cis-acting elements, gene structure, and extracted transcriptome expression data, revealed that the majority of genes within this family are implicated in the response to, and regulation of, both biotic and abiotic stresses. Further real-time fluorescence quantitative PCR analysis revealed that certain genes within this gene family respond to both biotic and abiotic stresses. The majority of this gene family is expressed on the cell membrane, exhibits sensitivity to external stress signals, and is involved in signal transduction. We believe that identifying and functionally analyzing this gene family in maize can yield new insights and solutions for research aimed at enhancing the stress tolerance of maize.

## 1. Introduction

Protein kinases are key regulatory molecules of cellular function. Receptor-like kinases are the most abundant protein kinase family in plants, accounting for approximately 60% of kinase reviews [1]. Hormone receptor-like kinase is an important part of the receptor-like kinase family, and its N-terminal condensin domain can reversibly bind to carbohydrates. LecRLK research primarily focuses on model plants, identifying 173 LecRLKs in rice and 75 in Arabidopsis thaliana [2,3]. In recent years, advancements in genome sequencing technology and in-depth studies of functional genomics have increasingly highlighted the role of LecRLKs in plant growth and development, as well as their responses to biotic and abiotic stresses [4].

Lectin receptor-like protein kinases represent a significant subfamily within the broader receptor-like protein kinase family. This subfamily is characterized by three primary domains: the extracellular lectin domain, the transmembrane domain, and the intracellular kinase domain. LecRLKs can be categorized into three distinct types based on their extracellular lectin domains: L, G, and C types [5]. The L-type lectin-like receptor kinase contains a lectin (legume-lectin) domain at its N-terminus, which enables it to detect external signals through hydrophobic interactions [6]. The G-type lectin receptor kinase, which contains a Galanthus nivalis agglutinin domain, exhibits an affinity for α-D mannitol [7]. C-type lectin receptor kinases, characterized by a calcium-dependent lectin domain, are rarely identified in plants, and their functional roles remain largely unclear [8]. Lectin receptor kinases are primarily situated on the cell membrane, where they recognize pathogens, symbionts, and stress signals. They play a crucial role in coordinating cell growth and are involved in plant development as well as innate immune responses to stress [9]. Condensin-like receptor kinases recognize specific ligands, initiate signaling pathways, detect environmental changes, and regulate the balance between growth and defense, thereby playing a crucial role in enhancing plant adaptability [10]. *ZmPK1* is the first lectin receptor kinase identified in a plant species. It comes from maize and belongs to G-type LecRLKs. Its extracellular domain is involved in the recognition between pollen and stigma [11].

Researchers have systematically identified LecRLKs associated with plant growth and development in Arabidopsis, rice, wheat, and various other crops [10]. Wan J et al. found that a mutation in LecRLK-IV.2 in Arabidopsis thaliana led to a reduction in pollen size [12]. Deng K et al. found that mutations in the LecRLK-V.2 gene of Arabidopsis thaliana result in decreased sensitivity of seeds to abscisic acid (ABA) [13]. Through research on the LecRLK-VIII.2 gene in Arabidopsis thaliana, Lin et al. discovered that it functions upstream of the MAPK gene to regulate yield [14]. Cheng X et al. investigated the G-type lectin receptor kinase OsLecRLK in rice, revealing that this gene can inhibit α-amylase activity in seeds and directly interacts with ADF [15]. Dharmateja P et al. identified the candidate gene TraesCS2D02G126300 (LecRLKs) through QTL mapping, which may play a role in regulating the phosphorus utilization efficiency of plants. This gene can be expressed in wheat roots under both normal and phosphorus-deficient conditions [16].

The LecRLKs family in plants also participates in biological stress responses. L-type lectin-like receptor kinases were identified in disease resistance studies of Arabidopsis, cucumber, wheat, barley (Hordeum vulgare), pepper, and apple (Malus domestica) [17]. Arabidopsis LecRK-I.9 is involved in regulating the jasmonic acid (JA) signaling pathway. Defective mutants are impaired in cell wall defense and JA signaling, and are resistant to Pseudomonas syringae, oomycete pathogens, Phytophthora brassicae and the sensitivity of Phytophthora capsici increased, and overexpression of the LecRK-I.9 gene enhanced the plant’s resistance to the pathogen [18]. LecRK-I.9 interacts with the effector protein of Phytophthora infestans, influencing the adhesion between the cell wall and the cell membrane, thereby regulating plant disease resistance [19]. The LecRK-I.5 and LecRK-I.9 receptors may exhibit a synergistic effect, as evidenced by the observation that double mutant plants show an exacerbated susceptibility to infection by Pseudomonas syringae [20]. Micol-Ponce et al. identified the L-type lectin receptor protein kinase LecRK-V.5 in Arabidopsis [21]. This gene plays a critical role in oligosaccharide and hormone signal transduction, as well as in mediating plant disease resistance responses [22]. Research indicates that L-type lectin receptor proteins primarily mediate plant responses through stomatal regulation and ATP signaling [23]. LecRK-V.5 mediates plant immune responses by negatively regulating stomatal function. Following the editing of LecRK-V.5, the resistance of plants to PstDC3000 will be significantly enhanced; conversely, the overexpression of this gene will render plants more susceptible to PstDC3000 [24].

LecRLKs are involved in plant responses to abiotic stress and resist abiotic stress, such as salt stress, low temperature stress, drought stress, and mechanical damage [25]. LecRLKs transmit signals to downstream pathways via autophosphorylation or the phosphorylation of other proteins [26]. SIT1 is activated in response to salt stress, phosphorylating downstream effectors (MAPK3/6), which triggers reactive oxygen species (ROS) accumulation induced by ethylene signals, thereby enhancing plant sensitivity [10]. Research on the LecRLK family genes in Arabidopsis, soybean, and tobacco has demonstrated that a majority of these genes respond to salt-alkali stress and mechanical damage, contributing to the development of specific resistance mechanisms [27].

Maize is an important food crop, accounting for 43% of global food production [28]. Additionally, maize serves as a significant source of feed for the animal husbandry and breeding industries, and it is also one of the essential raw materials for the light and chemical industries. However, maize yields are frequently impacted by drought, salinity, and leaf spot diseases, leading to substantial yield losses [29]. Given the role of LecRLK family genes in enhancing plant tolerance to both biotic and abiotic stress, it is essential to identify and investigate these genes in maize to improve the maize’s resilience to stress conditions [30]. This study presents a systematic review of the maize LecRLK gene family, analyzing their physical and chemical properties, sequence characteristics, phylogenetic relationships, promoter cis-acting elements, and gene expression patterns. This comprehensive analysis lays the groundwork for future research into the biological functions of maize LecRLK genes. This is the first comprehensive research report on maize LecRLKs, filling the gap in the identification and functional analysis of maize LecRLKs.

## 2. Materials and Methods

### 2.1. Plant Materials

The plant material utilized in this study was the B73 inbred line of corn, which was supplied by the Institute of Agricultural Biotechnology at the Jilin Academy of Agricultural Sciences. Plants were potted and subsequently placed in an artificial climate chamber, which was maintained at a temperature of 25 °C and a humidity level of 70% for cultivation. The potting soil is a 1:1 mixture of nutrient soil and vermiculite. During the cultivation period, the plants were watered every three days until they reached the V5/V6 developmental stage, after which stress treatment was administered. For conditions involving drought stress, combined drought and salt stress, gray leaf spot stress, and anthracnose stress, please refer to the work of Cristian and Swart [31,32].

### 2.2. Identification and Sequence Analysis of Maize Lectin Receptor Proteins

Arabidopsis serves as an excellent reference for gene family studies owing to its small genome size, high-quality annotations, extensive functional studies, and its status as a model organism in plant genetics. Utilizing Arabidopsis thaliana allows us to leverage existing data for the identification of both conserved and novel genes in maize through a comparative genomics approach. Maize genome data and GFF annotation information were extracted from the Ensembl database (http://plants.ensembl.org/index.html) (accessed on 2 March 2024) and the Arabidopsis Information Website (TAIR) (https://www.arabidopsis.org/). Additionally, the amino acid sequence of the Arabidopsis LecRLK gene was downloaded. We utilized the Blast Compare Two Seqs module in TBtools v2.101, setting the E value to 10^−5^ for local BLAST analysis. Additionally, the Pfam database (http://pfam.xfam.org/) was employed to identify the characteristic domain of the maize LecRLK gene family (PF00139, PF01453, PF00059), while the amino acid sequence of the maize LecRLK family was screened using the Simple HMM Search. To analyze the sequences, we first took the intersection of the sequence IDs from the two screenings. Then, we utilized the online bioinformatics tools provided by the National Center for Biotechnology Information (NCBI) Conserved Domain Database (CDD) at https://www.ncbi.nlm.nih.gov/Structure/bwrpsb/bwrpsb.cgi (accessed on 2 March 2024) and the SMART online website at https://smart.embl.de/smart/set_mode.cgi?NORMAL=1 (accessed on 3 March 2024) to determine whether the sequences contain the LecRLK conserved domain. Next, CD-hit- v4.6.8-2017-1208 software (https://github.com/weizhongli/cdhit/releases/download/V4.6.8/cd-hit-v4.6.8-2017-1208-source.tar.gz) was employed to retain only those sequences exhibiting a minimum of 90% sequence homology to known Arabidopsis genes, thereby ensuring a high-confidence identification of homologous genes. This threshold was selected as it strikes a balance between sensitivity, which involves capturing closely related sequences, and specificity, which aims to minimize false positives. Finally, different transcripts of the same gene were manually removed, ensuring that only the longest transcript sequence was retained [33].

### 2.3. Physical and Chemical Property Analysis and Chromosome Localization

The ProtParam tool (https://web.expasy.org/protparam/) (accessed on 5 March 2024) was utilized to calculate the predicted physicochemical characteristics of the maize LecRLK gene protein. This analysis included the determination of the amino acid sequence length, theoretical isoelectric point (pI), molecular weight (Mw), instability index, fatty index, and other relevant indicators. The CELLO online website (http://cello.life.nctu.edu.tw/) (accessed on 9 March 2024) was utilized to predict the subcellular localization of the maize LecRLK gene. The chromosome location information for the maize LecRLK gene was extracted from the maize genome annotation file and subsequently visualized using the Gene Location Visualize from GTF/GFF module in Tbtools software v2.1 (https://github.com/CJ-Chen/TBtools/releases, accessed on 8 September 2023) [34].

### 2.4. Phylogenetic, Gene Structure, and Conserved Sequence Analysis of LecRLK Gene in Maize

The amino acid sequences of the *LecRLK* genes from maize and Arabidopsis were chosen for the analysis of their phylogenetic relationships. Multiple sequence alignment was conducted using Clustal. Upon completion of the alignment, the Neighbor-Joining (NJ) method implemented in MEGA 11 software (https://www.megasoftware.net/) was utilized to construct a phylogenetic tree. Since the construction of the evolutionary tree relies on model calculations, the accuracy of each branch node necessitates 1000 bootstrap replicates remain to be determined. Therefore, the Bootstrap parameter is set to 1000. Subsequently, iTOL (https://itol.embl.de/) (accessed on 9 May 2024) was employed to visualize and enhance the aesthetics of the phylogenetic tree.

The Online MEME software (https://meme-suite.org/meme/) (accessed on 9 May 2024) was utilized to analyze the motif structure of the maize LecRLK family members. In this analysis, the maximum number of motifs was set to 20, the minimum width of the motifs was set to 6, and the maximum width of the sequences was set to 50. The structure of the maize LecRLK gene, encompassing both the coding sequence (CDS) and untranslated regions (UTR), was extracted from the maize genome annotation file. Subsequently, TBtools v2.101 was employed for visual analysis [35].

### 2.5. Collinearity Analysis and Gene Duplication of Maize LecRLK Gene

The BLASTP program was used to identify homologous *LecRLK* genes in maize, with the e-value threshold set to <e^−5^. The collinear relationship between maize *LecRLK* genes was analyzed using MCScanX default parameters. TBtools was used for visualization.

The collinearity comparison with Arabidopsis, rice, and sorghum is of great significance. Arabidopsis thaliana is a widely used model organism in the fields of plant genetics and molecular biology. Rice is a typical monocotyledonous model plant. The genome data of both species are complete and their gene functions have been extensively identified. Furthermore, rice, sorghum, and corn are members of the same family of grass crops, and their homologous genes are relatively conserved. The inclusion of these organisms facilitates a robust collinearity analysis, which is essential for studies of gene families. Collinearity refers to the consistency of gene sequences across different species or within a single genome, serving as evidence for evolutionary relationships, including gene duplication, divergence, and retention. By analyzing collinearity, we can trace the evolution of gene families, identify common ancestors, and detect species-specific adaptations. This comparative approach enhances our understanding of the evolutionary mechanisms of plant genes and their functional diversity. For the collinearity analysis, genome sequence and annotation files for Arabidopsis and rice were downloaded from the Phytozome v13 website (https://phytozome-next.jgi.doe.gov/) (accessed on 13 May 2024), while the sorghum genome sequence and annotation files were obtained from the NCBI website (https://www.ncbi.nlm.nih.gov/) (accessed on 15 May 2024). The MCScanX program was then utilized to analyze the collinearity between maize and Arabidopsis, as well as between rice and sorghum [36].

### 2.6. Cis-Acting Elements and miRNA Prediction

To identify potential regulatory mechanisms of gene expression, we analyzed the 2000 bp upstream region of candidate genes, which is commonly referred to as the promoter region. This region was selected due to its frequent presence of cis-regulatory elements (CREs), which are crucial for regulating transcriptional activity. Research has demonstrated that cis-regulatory elements (CREs) located within 2000 base pairs upstream of the transcription start site (TSS) play a crucial role in mediating gene responses to developmental signals and environmental stimuli. Promoter sequences were extracted using TBtools and subsequently analyzed with PlantCARE (https://bioinformatics.PSB.ugent.be/webtools/plantcare/html/) (accessed on 20 May 2024) to predict cis-regulatory elements. The HeatMap module in TBtools was employed to visualize and summarize the cis-elements associated with stress response, plant growth and development, plant hormone response, and light response.

The miRNA targeting *ZmLecRLK* was predicted using the psRNATarget website (https://www.zhaolab.org/psRNATarget/analysis?function=3, accessed on 5 September 2023) with the default parameters [37]. Perfect or near-perfect complementarity between the miRNA and target mRNA sequences is the primary criterion for predicting and screening RNAs, particularly within the 5′ untranslated region (5′ UTR). This region, comprising 2 to 8 nucleotides at the 5′ end of the miRNA, is critical for target recognition.

### 2.7. RNA Seq Data and Expression Pattern Analysis of LecRLK

To investigate the expression profiles of LecRK family genes, we selected publicly available RNA-seq datasets from repositories such as NCBI SRA and GEO. The datasets were chosen based on the following criteria: To achieve a comprehensive analysis of gene expression across different genetic backgrounds, it is essential to include maize tissues such as roots, stems, leaves, and flowers, as well as stress conditions including salt stress, drought stress, and leaf spot. Data quality control involves the inclusion of only high-quality RNA-seq datasets. Raw reads are assessed using tools such as FastQC, which evaluates quality parameters including base call accuracy, GC content, and nucleic acid contamination. This study utilized three maize RNA-seq datasets: (1) drought and salt stress responses in B73 and its mutants (NCBI SRA PRJNA1002756) (https://www.ncbi.nlm.nih.gov/sra/?term=, accessed on 8 September 2023); (2) expression levels of corn B73 across 19 tissue types (NCBI SRA PRJNA1002756); and (3) expression levels of leaves following infection by Colletotrichum corneum and Cercospora zeae (NCBI SRA PRJNA 977728). Data quality control was performed by Fastq. According to the quality control results, data filtering was performed using Sickle software (http://www.bio-trainee.com/jmzeng/sickle/sickle-result) to remove low-quality sequences and ensure paired-end symmetry.

TopHat2 software (http://ccb.jhu.edu/software/tophat/downloads/tophat-2.1.0) was used for transcriptome-genome alignment, and BowTie2 software (http://bowtie-bio.sourceforge.net/bowtie2) was used to build an index with Zm-B73-REFERENCE-V4 as the reference genome before alignment. The comparison results were counted by ht-seq, and the count value of the different experimental groups was finally output [31].

### 2.8. RNA Extraction and qRT PCR

The total RNA was isolated from 100mg plant tissues following a TRIzol method, while the quantity and quality of RNA was determined using a NanoDrop spectrophotometer.

The cDNA reverse transcript was produced from RNA using a Thermo cDNA kit ((Shanghai, China)). qRT-PCR primers were designed using IDT online software (https://sg.idtdna.com/pages/tools), the internal reference gene was Actin II, and the primers were synthesized by Kumei Biological Co., Ltd. The reactions were completed in a 25-μL total volume using the SYBR Green PCR Master Mix kit and a Light Cycler^®^480II Sequence Detection System. The relative expression levels of candidate genes were calculated by comparison using the 2^−∆∆ct^ method. The qRT-PCR and data analysis were performed using methods described by He et al. [38].

## 3. Results

### 3.1. Identification and Phylogenetic Analysis of LecRLK Gene Family Members in Maize

By comparing with the LecRLK family genes of Arabidopsis thaliana, 35 L-type genes, 54 G-type genes, and 1 C-type gene of the LecRLK family genes were identified in the maize genome. The L-type unique legume-lectin domain (Lectin_legB PF00139), the G-type unique Galanthus nivalis agglutinin (lectin_B PF01453) domain, and the C-type unique C-type (Lectin_C PF00059) domain. The identified L-type and G-type LecRLK family genes both possess STK kinase domains and transmembrane domains; however, type C (*Zm00001d040192*) lacks these features, leading to its classification as a uncomplete gene. In the evolutionary tree co-constructed with the LecRLK family genes in Arabidopsis thaliana, the L type and G type were classified into four subgroups based on evolutionary distance, designated as Group I, Group II, Group III, and Group IV, respectively. Within the G type subgroup, Group I comprises 7 *LecRLK* genes, Group II contains 2 *LecRLK* genes, Group III includes 4 *LecRLK* genes, and Group IV consists of 42 *LecRLK* genes (Figure 1).

### 3.2. Analysis of ZmLecRLK Gene Structure and Conserved Domain

Analysis of the gene structure of the *ZmLecRLK* gene family shows that the ZmLecRK family genes all contain transmembrane domains and kinase domains (Figure 2E). The difference is that the G type contains the Galanthus nivalis agglutinin domain and the L type contains the legume-lectin domain (Figure 2E). Ten motifs predicted by MEME were identified within the *ZmLecRLK* gene family, with the majority containing motifs 1 through 8, whereas the G type encompasses all motifs (Figure 2B). Notably, motifs 9 and 10 appear to be exclusive to the agglutinin domain of Galanthus nivalis (Figure 2B,D).

Analysis of the exon-intron structure of the *ZmLecRLK* gene found that the number of exons ranged from 1 to 11 (Figure 2C), of which 59 genes had only one exon, and Zm00001d002172 contained 11 exons. The majority of L-type *ZmLecRLK* genes (82.35%) contain one to two exons, whereas genes with six to nine exons are predominantly found among G-type genes (87.5%).

### 3.3. Chromosome Localization and Replication Events of ZmLecRK Gene Family

The 89 LecRK family genes identified in this study are unevenly distributed across the 10 chromosomes of maize. Chromosome 1 contains the highest number, with 14 LecRK genes, followed by chromosome 10 with 13 LecRK genes and chromosome 7 with 12 LecRK genes. In contrast, only three LecRK family genes are present among the remaining seven chromosomes (Figure 3).

We conducted a gene duplication analysis of the LecRK gene family using BLAST and MCScanX. Our analysis identified five pairs of tandemly repeated genes: *Zm00001d032868* and *Zm00001d032869*, *Zm00001d021726* and *Zm00001d021727*, *Zm00001d037397* and *Zm00001d037398*, *Zm00001d025390* and *Zm00001d025391*, as well as *Zm00001d025393* and *Zm00001d025394*, which are located on chromosomes 1, 6, 7, and 10, respectively (Figure 3). Gene duplication frequently results in genetic mutations that give rise to new functions, significantly contributing to plant adaptation to environmental changes. To investigate the gene duplication events within the *ZmLecRLK* gene family, a Circos map was constructed. This analysis reveals the presence of 13 *ZmLecRLK* gene pairs in maize, which aligns with the occurrence of whole genome duplication (WGD) in this species (Figure 4A).

To further analyze the evolutionary relationships between maize and other species, we compared the collinearity of the LecRK genes in maize with those in three other representative species: Arabidopsis, rice, and sorghum (Figure 4B). The results indicated that there were 4, 55, and 62 pairs of collinear genes between corn and the three respective plants. Chromosomes 1 and 4 of Arabidopsis exhibit collinearity with chromosomes 1, 3, and 7 of maize, respectively. Additionally, the collinear gene pair of rice LecRK is located on all 10 chromosomes of maize. The collinear genes of sorghum LecRK are aligned with the 10 chromosomes of maize, with the exception of chromosome 5. The LecRK family genes are less conserved between dicotyledonous and monocotyledonous plants, but more conserved in rice and sorghum, which are both gramineous crops (Figure 4B).

### 3.4. ZmLecRLK Protein Characteristics and Subcellular Localization Prediction

In terms of physical and chemical properties, the length of *ZmLecRLK* proteins varies from 398 to 1347 amino acids. The longest protein, *Zm00001d024637*, consists of 1347 amino acid residues, followed by *Zm00001d010971*, which contains 1193 amino acid residues. The smallest protein identified is *Zm00001d043257*, which comprises 398 amino acid residues (Appendix A). The estimated molecular weight of the *ZmLecRLK* protein ranges from 44.5 to 150.3 kDa (Appendix A). The hydrophilicity analysis indicated that all *ZmLecRLK* proteins are classified as hydrophilic (Appendix A). Prediction of subcellular localization of the *ZmLecRLK* gene family shows that the expression location of this family of genes is relatively wide, with 67 genes distributed on the cell membrane, 63 of which are also located in the cytoplasm, and 7 genes are also located outside the cell. Interestingly, two genes are located in the nucleus, which is different from most *ZmLecRLK* family genes.

### 3.5. Cis Acting Elements and miRNA Prediction Size

We predicted the cis-acting elements located 2000 bp upstream of each *ZmLecRLK* promoter region in maize and mapped them by excluding common elements such as the TATA box and CAAT box (Figure 5A). The 604 identified cis-acting elements were categorized into four groups: development-related elements, environmental stress-related elements, hormone-responsive elements, and light-responsive elements (Figure 5B,C). The primary hormone response elements include the abscisic acid response element (ABRE), the methyl jasmonate response elements (TGACG motif and CGTCA motif), and the gibberellin response elements (P-box and GARE motif). The second largest category comprises photoresponsive elements, which include I-box motifs, G-box elements, and Sp1 originals. The third category consists of environmental stress-related elements, encompassing anaerobic inducible elements (ARE), low temperature response elements (LTR), and enhancer-like elements that are involved in hypoxia-specific induction. The fourth category is developmental related elements, including endosperm expression elements (GCN4_comotif), cis regulatory elements involved in endosperm specific negative expression elements (CAT box), and maize soluble protein metabolism regulation (CCAAT box).

MicroRNAs (miRNAs) serve as crucial tools for post-transcriptional regulation by binding to the 5′ untranslated region (5′UTR) of genes, thereby modulating the translation process. Through the analysis of gene structures, it is evident that among the 89 identified genes, 14 possess 5′ UTR regions. The prediction of miRNAs bound to these 5′ UTR regions indicates that *Zm00001d021434*, *Zm00001d021727*, *Zm00001d032408*, *Zm00001d043253*, and *Zm00001d018182* exhibit specific miRNA binding within their 5′ UTR regions (Table 1). Although there are miRNAs located in the 5′ UTR regions of these five genes, their functions differ. Specifically, only *Zm00001d021434* and *Zm00001d032408* inhibit translation, while the miRNAs associated with the other genes serve to block expression through different mechanisms (Table 1). Nevertheless, all of these miRNAs inhibit the normal expression of their respective target genes.

### 3.6. GO En3.6 Richment Analysis of ZmLecRLK

The Gene Ontology (GO) annotation of the *ZmLecRLK* gene family is categorized into three aspects: molecular function, cellular component, and biological process. The enrichment of annotated functions indicates that the molecular functions of the *ZmLecRLK* family genes are primarily associated with protein kinase activity, phosphotransferase enzyme activity, protein serine/threonine kinase activity, transmembrane receptor protein serine/threonine kinase activity, transmembrane receptor protein kinase activity, transmembrane signaling receptor activity, and molecular sensor activity (Figure 6). In terms of biological processes, the focus is primarily on the recognition of pollen, the enzyme-linked receptor protein signaling pathway, the transmembrane receptor protein serine/threonine kinase signaling pathway, and the defense responses to oomycetes and bacteria, as well as interactions with other organisms. Additionally, the body’s defense response encompasses the biological processes involved in interactions between species and the regulation of these processes (Figure 6). Regarding cellular components, the emphasis is mainly on the plasma membrane and cell periphery.

### 3.7. RNA-seq analysis Analysis of ZmLecRLK

To investigate the function of *ZmLecRLK* family genes, we examined publicly available RNA-seq expression data, focusing on the expression patterns of L-type and G-type genes across various maize tissue types, as well as under biotic and abiotic stress conditions. Based on this analysis, a heat map of gene expression was generated, revealing that the majority of *ZmLecRLK* genes are predominantly expressed in leaves, particularly concentrated in the canopy region. In the context of abiotic stress, the majority of L-type *ZmLecRLK* genes exhibit responses to salt stress, drought stress, and mixed stress conditions (Figure 7A–D). Notably, the responses of the *ZmLecRLK* family genes to these stresses are markedly distinct. For instance, genes such as *Zm00001d025920*, *Zm00001d019334*, *Zm00001d006627*, and *Zm00001d037397* demonstrate significant expression in drought-tolerant mutants under drought stress, whereas they are nearly undetectable in the B73 genotype (Figure 7B). Genes such as *Zm00001d037371*, *Zm00001d035476*, and *Zm00001d030759* exhibit higher expression levels in the non-stress environment of B73 and play a negative regulatory role in drought-tolerant mutants (Figure 7B). A similar pattern is observed with the G-type *ZmLecRLK* gene. Specifically, a total of 17 genes, including *Zm00001d021737*, demonstrate an active response to drought and salt stress in B73, characterized by increased expression (Figure 7A). In contrast, these genes are negatively regulated and show reduced expression in drought-tolerant mutants. In other G-type genes, the opposite is true.

In biotic stress, *ZmLecRLK* family genes also show different expression patterns. Analysis of the *ZmLecRLK* gene family expression under the stress of Cercospora zeae and Colletotrichum infection revealed that the L-type genes *Zm00001d019334*, *Zm00001d045838*, *Zm00001d043781*, and *Zm00001d037398*, among others, responded to Cercospora zeae infection (Figure 7E). Notably, the expression of these genes was localized to the leaves above the ear. Other L-type genes, including 12 genes such as *Zm00001d051615* and *Zm00001d053322*, respond to anthrax infection and the response time primarily occurs between 24 and 48 h post-infection (Figure 7E). The G-type *ZmLecRLK* gene exhibits behavior akin to that of the L-type under biotic stress; however, a key distinction is that the G-type *ZmLecRLK* gene encompasses genes that respond concurrently to infections caused by Cercospora zeae and Colletotrichum (Figure 7F). The G-type *ZmLecRLK* gene exhibits behavior akin to that of the L-type under biotic stress; however, a notable distinction is that the G-type *ZmLecRLK* gene encompasses genes that respond concurrently to infections caused by Cercospora zeae and Colletotrichum (Zm00001d016223, Zm00001d025035, Zm00001d031377, Zm00001d006297, Zm00001d021737) (Figure 7F).

### 3.8. Expression Level Analysis of ZmLecRLK

RNA-seq analysis of LecRLK family genes in response to biotic and abiotic stress indicates that the majority of *LecRLK* genes are capable of responding to both types of stress. Consequently, we selected five genes that exhibited responses to both biotic and abiotic stresses, and we quantified their expression in maize using quantitative PCR (qPCR) under various stress conditions. The results are presented in the Figure 8. Under drought stress conditions, *Zm00001d021737*, *Zm00001d031377*, and *Zm00001d019411* exhibited a positive response after 10 days of exposure to drought stress (Figure 8). However, *Zm00001d002536* was expressed at a significantly higher level seven days post-rehydration. Under the combined stress of drought and salinity, *Zm00001d019411* exhibited a highly significant expression after 10 days of stress and continued to show significant expression following seven days of rehydration (Figure 8). In contrast, *Zm00001021737* demonstrated no response to stress but was highly expressed after seven days of rehydration (Figure 8).

Under typical conditions of corn gray leaf spot disease, *Zm00001d021737* and *Zm00001d031377* are expressed at significantly higher levels in the leaves beneath the ear (Figure 8). In response to Colletotrichum stress, both *Zm00001d021737* and *Zm00001d031377* exhibited an immediate reaction to the stress. *Zm00001d002536* and *Zm00001d019411* exhibited highly significant expression 48 h post-infection, whereas *Zm00001d010700* demonstrated a significant increase in expression 24 h after infection (Figure 8).

## 4. Discussion

*ZmLecRLK* is a significant branch of plant kinases that not only regulates growth and development but also plays a crucial role in responding to both biotic and abiotic stresses [39]. Although 75, 173, 38, 22, and 185 *LecRLK* family genes have been identified in Arabidopsis, rice, tobacco, tomato, and soybean, respectively, there have been no reports on the *ZmLecRLK* gene family in maize to date [3,40,41]. Research on the *ZmLecRLK* gene in maize has primarily focused on *ZmPK1* [11]. In this study, we identified 89 *LecRLK* genes based on their similarity to the Arabidopsis LecRLK proteins [3]. We constructed an evolutionary tree for this gene family and conducted analyses of conserved domains, chromosomal locations, collinearity, and expression quantification. The structural characteristics of the LecRLK gene family include a kinase domain, a transmembrane domain, and either an L-type or G-type domain. Based on these conserved domains, the gene family is classified into L-type and G-type subfamilies. In the phylogenetic analysis of the *LecRLK* genes in Arabidopsis thaliana and maize, four subfamilies (L-I~IV, G-I~IV) were identified for each species (Figure 1). This finding is consistent with the results of the phylogenetic analysis conducted on rice. Members of the same family typically exhibit similar functions. For instance, *At5g01540*, *At5g01550*, and *At5g01560* are all implicated in seed germination and ABA response [13,19,42]. Consequently, *Zm00001d043781*, which resides on the same evolutionary branch as these genes, may also share a similar function in maize (Figure 1). *At5g65600* responds to biotic stress by phosphorylating AvrPtoB [43]. The genes *Zm00001d012625*, *Zm00001d018059* and *Zm00001d023253*, which are part of the same evolutionary branch as *At5g65600*, exhibit increased expression in response to gray leaf spot and anthracnose stress. Therefore, it is likely that *At5g65600* employs a similar response mechanism (Figure 1).

The chromosomal location of the *ZmLecRLK* gene family indicates that all members are randomly distributed across 10 chromosomes, comprising a total of 11 gene clusters, which include five pairs of tandem duplications (Figure 4). It is widely acknowledged that segmental duplications, tandem duplications, and translocation events are the primary drivers of gene family expansion. Segment duplication events of homologous genes typically occur in distant regions, while tandem duplications take place in adjacent regions. In our study, two of the five pairs of tandem duplications were identified in the G-IV clade, while two pairs were found in the L-IV clade (Figure 4A). This indicates that the L and G subtype IV clades are the primary branches in which tandem duplications occur. A substantial body of evidence indicates that variations in promoter regions are frequently associated with gene activity, and that transient elements within these regions play a significant role in regulating gene expression. We categorized the promoters associated with the *ZmLecRLK* gene family into four distinct types: those related to development, environmental stress, hormone response, and light response (Figure 5). In addition to light-responsive elements, which are relatively conventional promoter elements, hormone-responsive elements and environmental stress elements constitute a significant proportion. Furthermore, most LecRLK proteins exhibit responses to both biotic and abiotic stresses. Abscisic acid (ABA) and gibberellins (GA) have been shown to play significant roles in plant responses to abiotic stresses, particularly drought, low temperatures, and salinity. Upon sensing these abiotic stresses, plants accumulate ABA and GA, which in turn promotes the expression of associated stress proteins and defense genes [44]. Our analysis revealed that 18 gene promoters contain cis-acting elements associated with abscisic acid (ABA) response, while 26 gene promoters are linked to gibberellin (GA) response. This suggests that these genes may play a crucial role in regulating abiotic stress defense mechanisms in maize. This suggests that these genes may play a crucial role in regulating abiotic stress defense mechanisms in maize. The genes that play an important role, especially those connected in series with cis-acting elements in response to ABA and GA, will be the focus of further research.

Gene expression profiles provide important insights into the functional roles of genes. To determine the expression pattern of *ZmLecRLK*, we further analyzed transcriptome data from various tissue types and stress conditions in maize. The results of this analysis indicated that the majority of *ZmLecRLK* transcripts were expressed in maize leaves, particularly concentrated in the canopy leaves (Figure 7). The *ZmLecRLK* gene is implicated in reproductive growth, particularly in the processes of cell division and leaf senescence. Zm00001d025920 has been shown to respond to drought and salt stress, while the rice gene Os02g0640500, which is collinear with it, has been confirmed to play a role in salt stress tolerance (Figure 7) [9]. Consequently, two collinear genes may exhibit functional similarities. In addition, we confirmed through quantitative PCR (qPCR) that there are genes within the *ZmLecRLK* family that respond to both biotic and abiotic stresses (Figure 8). Consequently, *ZmLecRLK*, as a receptor protein kinase possessing a transmembrane domain, is directly and extensively involved in the plant’s response to these stresses. Further exploration and validation of the functions of *ZmLecRLK* family genes will yield significant advancements in maize stress resistance research.

## 5. Conclusions

A total of 89 *LecRLK* genes were identified in maize and subjected to a series of bioinformatics analyses. Phylogenetic analysis reveals that both L-type and G-type *LecRLK* genes have four distinct branches. Notably, the gene structure of G-type LecRLKs is more complex, exhibiting a greater number of open reading frames (ORFs) and motifs. The promoter region of LecRLK contains cis-acting elements associated with processes of plant growth and development, as well as responses to environmental stress and hormones. Furthermore, analysis of gene expression patterns indicates that *LecRLK* genes respond to a range of biotic and abiotic stresses. These genes are predominantly expressed in canopy leaves, with varying expression levels, which will provide valuable insights for our future research endeavors.

## Figures and Tables

**Figure 1 biology-14-00020-f001:**
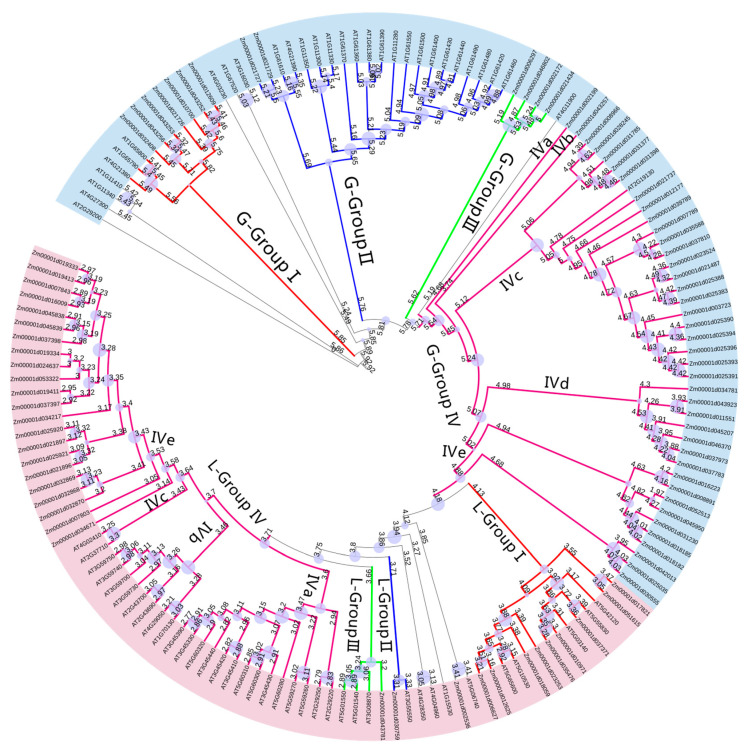
LecRLK protein phylogenetic tree. Zm represents the maize gene, and At represents the Arabidopsis gene. Pink represents L-type LecRLK protein, and blue represents G-type LecRLK protein. L-groupI~L-groupIV; are subfamilies of L-type LecRLK proteins, and G-groupI~G-groupIV are subfamilies of G-type LecRLK proteins.

**Figure 2 biology-14-00020-f002:**
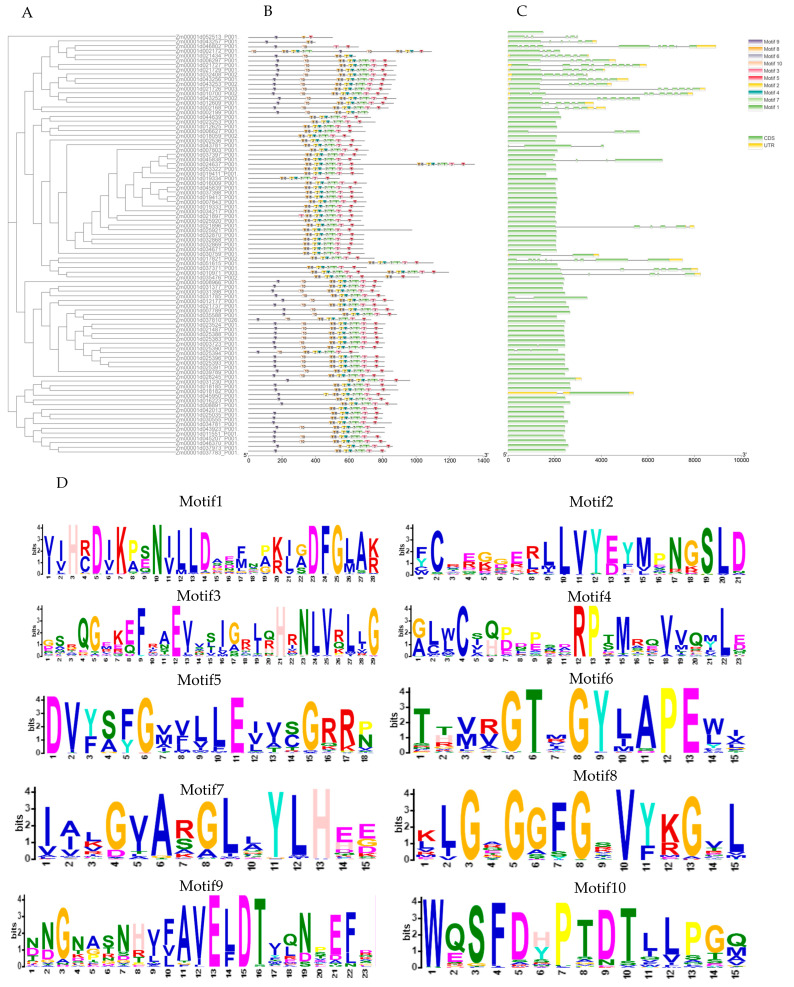
(**A**) Phylogenetic tree of maize *LecRLK* genes. (**B**) Conserved motifs of *ZmLecRLK* proteins. The 10 conserved motifs are represented by different colors. (**C**) Exon-intron structure of the *ZmLecRLK* gene. The green box indicates the CDS region of the gene, the orange box indicates the UTR region, and the black line indicates the intron. (**D**) A conserved motif of the maize LecRLK gene. Motif1-Motif10 represent different conserved motifs, the numbers on the x-axis represent the amino acid position, and the font size represents the relative frequency at the position. (**E**) Conservative domain of *ZmLecRLK* protein. Different conservative domains are represented by different colors.

**Figure 3 biology-14-00020-f003:**
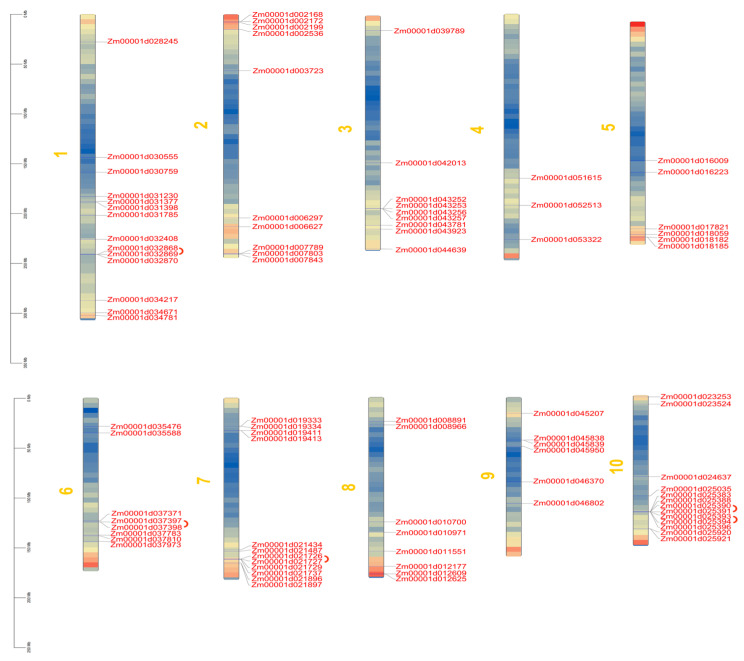
Chromosome distribution of *ZmLecRLK* gene. Set the sliding window size to 100 kb, with red to blue representing the gene density from high to low.

**Figure 4 biology-14-00020-f004:**
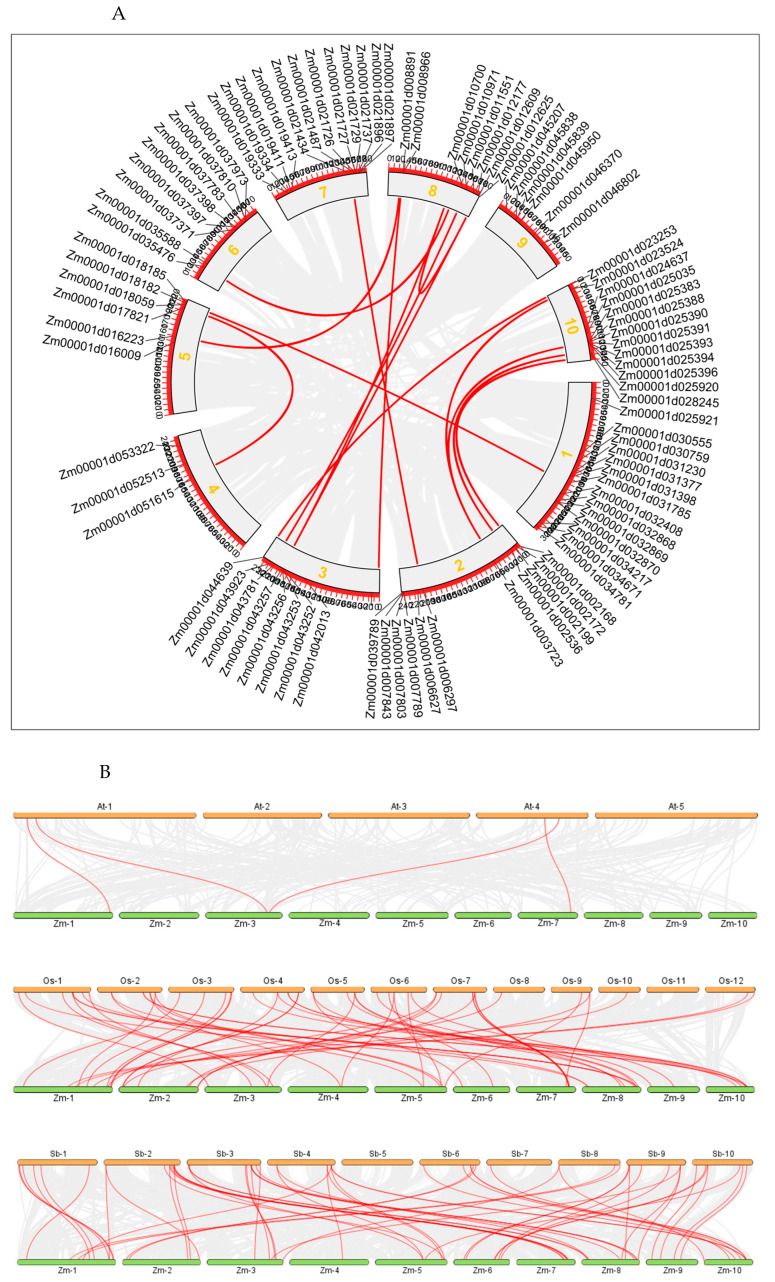
*ZmLecRLK* collinearity analysis. (**A**) Analysis of collinearity between *ZmLecRLK* genes. (**B**) The LecRK gene in maize is collinear with Arabidopsis, rice, and sorghum, respectively. The gray lines represent all collinear gene pairs, and the colored lines represent collinear *ZmLecRLK* gene pairs.

**Figure 5 biology-14-00020-f005:**
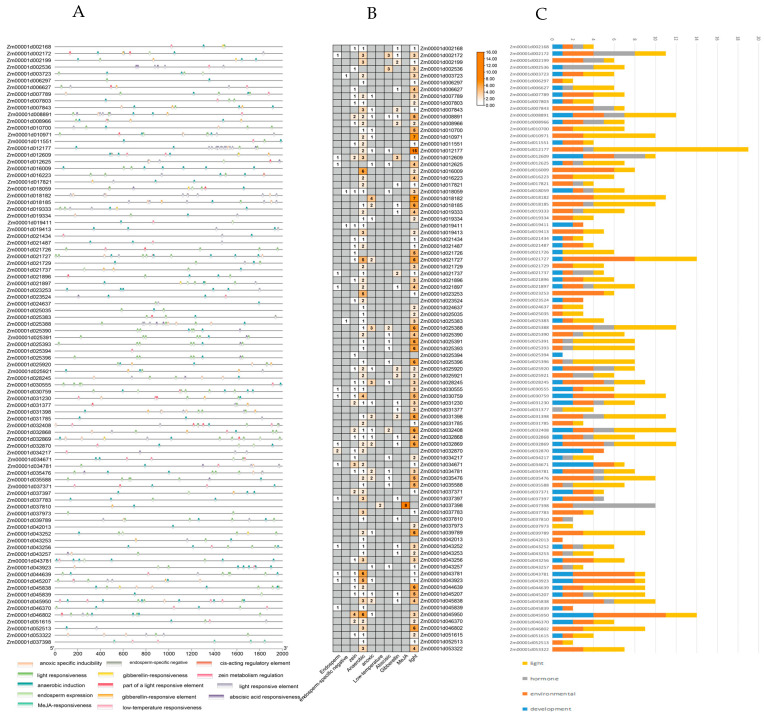
Analysis of cis-regulatory elements of maize *ZmLecRLK* gene. (**A**) Distribution of cis-acting elements in the promoter region of *ZmLecRLK* gene. (**B**,**C**) *ZmLecRLK* gene promoter number statistics. Yellow represents light response elements, gray represents hormone response elements, orange represents environmental stress related elements and blue represents development related elements.

**Figure 6 biology-14-00020-f006:**
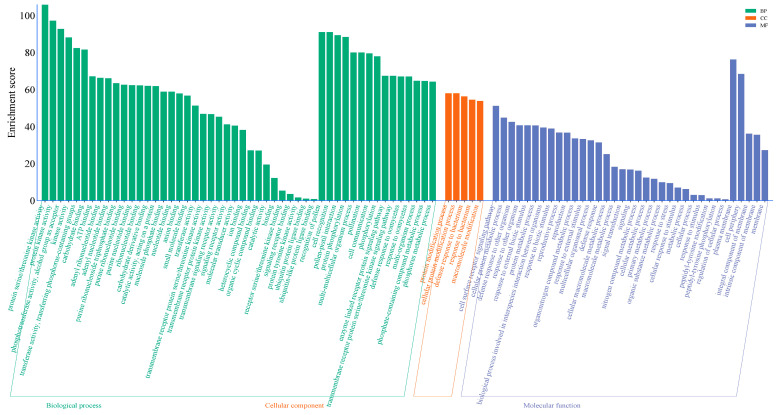
GO enrichment analysis of *LuMADS* genes.

**Figure 7 biology-14-00020-f007:**
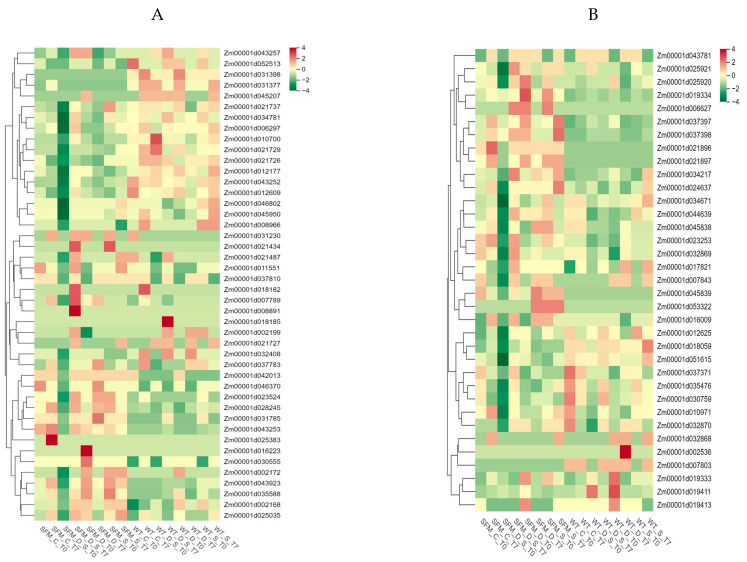
The expression patterns of *ZmLecRLK* genes. (**A**) Expression pattern of G type *ZmLecRLK* genes after salt treatment, drought treatment and drought -salt treatment. (**B**) Expression pattern of L type *ZmLecRLK* genes after salt treatment, drought treatment and drought -salt treatment. (**C**) The expression level of G type of *ZmLecRLK* in different tissues of maize. (**D**) The expression level of G type of *ZmLecRLK* in different tissues of maize. (**E**) Expression pattern of L type *ZmLecRLK* genes under Cercospora zeae and Colletotrichum stress. (**F**) Expression pattern of G type *ZmLecRLK* genes under Cercospora zeae and Colletotrichum stress.

**Figure 8 biology-14-00020-f008:**
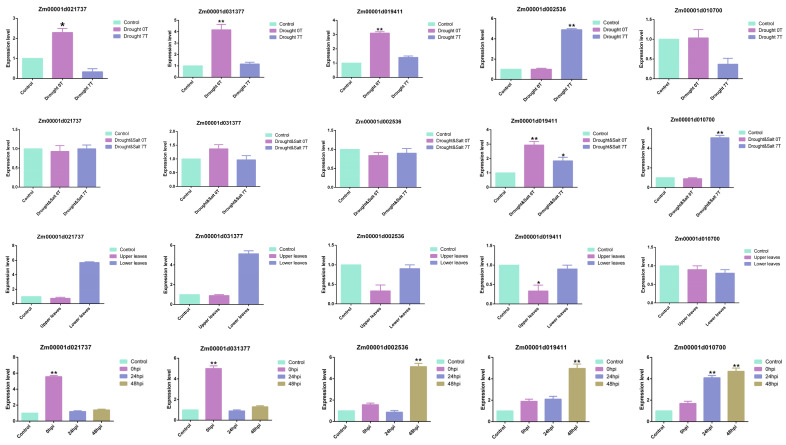
Analysis of expression pattern of *ZmLecRLK* gene in drought stress, drought-salt stress, Cercospora zeae and Colletotrichum stress response. Using Student’s *t*-test, asterisks indicate statistically significant differences (* *p* < 0.05; ** *p* < 0.01). ns represents no significance. Data are shown as mean ± SD from three independent experiments. 0T represents the expression level after 10 days of stress. 7T represents the gene expression level after 7 days of rehydration. Upper leaves represent the expression level of the target gene on leaves that grow above the female ear after being subjected to gray spot disease stress. Lower leaves refers to the expression level of the target gene on leaves that grow below the female ear after being subjected to gray spot disease stress. 0 hpi, 24 hpi, and 48 hpi represent the 0 h, 24 h, and 48 h of Colletotrichum infection, respectively.

**Table 1 biology-14-00020-t001:** Potential miRNA targets of *LuGRF* gene.

miRNA	Target	Expectation	miRNA Length	Target_Start	Target_End	Inhibition	Multiplicity
zma-miR399a/b/c/d/h-3p	Zm00001d021434	5	21	131	151	Translation	1
zma-miR160b/c/g-3p	Zm00001d021727/Zm00001d032408	4.5	21	201	221	Cleavage	1
zma-miR164a/b/c/d/e/f/g/h-5p	Zm00001d032408	5	21	165	184	Translation	1
zma-miR529-5p	Zm00001d032408	5	21	29	49	Cleavage	1
zma-miR1432-3p	Zm00001d043253	5	21	286	306	Cleavage	1
zma-miR2275b/c-3p	Zm00001d018182	5	21	782	803	Cleavage	1
zma-miR399a/b/c/d-5p	Zm00001d018182	5	21	555	575	Cleavage	1

## Data Availability

All data are reported in the article and Appendix A.

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
