# Peer review of "Identification and Characterization of the LecRLKs Gene Family in Maize, and Its Role Under Biotic and Abiotic Stress"

_biology, 2024, doi:10.3390/biology14010020_

Round 1
Reviewer 1 Report
Comments and Suggestions for Authors
The study investigates the LecRLK gene family in maize, revealing its potential role for stress resistance and growth regulation. Please improve manuscript on following points:
Line#28- Please change "for the development of enhanced maize line". enhanced for what?
See the text highlighted in text (attachment)
Line#292- Figure 4. Chromosome distribution of ZmLecRK gene. Set the sliding window size to 100 kb, with red to blue representing the gene density from high to low. Please elaborate the A and B in details for figure.

Author Response
Comments1:[Line#28- Please change "for the development of enhanced maize line". enhanced for what?]
Response1:[We agree with this comment, therefore, we have made corresponding modifications in the original text location.]
Comments2:[See the text highlighted in text (attachment)]
Response2:[We agree with this comment, therefore, we have made corresponding modifications in the marked areas.]
Comments3:[Line#292- Figure 4. Chromosome distribution of ZmLecRK gene. Set the sliding window size to 100 kb, with red to blue representing the gene density from high to low. Please elaborate the A and B in details for figure.]
Response3:[We agree with this comment, therefore,We have made modifications to the caption of Figure 4 as required.]
Reviewer 2 Report
Comments and Suggestions for Authors
LecRLK is a significant subfamily within the kinase family, playing a crucial role in the coordinated regulation of various growth and develop mental processes, as well as defense responses. However, the environmental defense mechanism of maize LecRLKs are largely unknown. In this study, 90 LecRLK genes were identified in the maize genome. And the gene expression was analyzed under salt, drought, and drought-salt treatments. The results provide evidence that LecRLKs are involved in growth, development and stress responses in maize, which lays a foundation for functional studies of ZmLecRLKs. Overall, the study is meaningful. However, the following questions are needed to be addressed:
1. While the English grammar in this article does not impede comprehension of its main ideas, there are a number of instances where certain word choices may not be entirely natural. To improve the article's readability for readers, it would be beneficial to have a native English speaker refine the language.
2. In the introduction part, the authors mention that “researchers have systematically identified LecRLKs associated with plant growth and development in Arabidopsis, rice, cotton, wheat, and various other crops." However, the details pertaining to cotton are absent from this part. I suggest the author add several sentences to enrich this point about cotton.
3. In figure 2, please add the bootstrap support on nodes in the ML tree and explain it with some sentences.
4. In the Results section, the figure legend of Figure 4 is wrong, please revise it carefully.
5. In figure 7-8, the name of these genes should be italicized. Please check this error in this manuscript.
6. In Figure 8, I don't think the author's illustration is very clear about the difference between control and drought 0T. In “Expression level analysis of ZmLecRLK”, "after 10 days of exposure to drought stress" was described. However, we have no find 10 days in Figure 8. Please check and amend it carefully.
7. Please check the format of references and amend it carefully.
Author Response
Comments1:[ While the English grammar in this article does not impede comprehension of its main ideas, there are a number of instances where certain word choices may not be entirely natural. To improve the article's readability for readers, it would be beneficial to have a native English speaker refine the language.]
Response1:[Thank you for your suggestion. We have optimized the language of the article accordingly.]
Comments2:[In the introduction part, the authors mention that “researchers have systematically identified LecRLKs associated with plant growth and development in Arabidopsis, rice, cotton, wheat, and various other crops." However, the details pertaining to cotton are absent from this part. I suggest the author add several sentences to enrich this point about cotton.]
Response2:[Thank you for your valuable feedback. In order to avoid disrupting the logic and structure of the article, we have removed cotton, so there is no need to add any further details related to cotton.]
Comments3:[In figure 2, please add the bootstrap support on nodes in the ML tree and explain it with some sentences.]
Response3:[We have redrawn Figure 2 according to your suggestion and made corresponding modifications.]
Comments4:[In the Results section, the figure legend of Figure 4 is wrong, please revise it carefully.]
Response4:[We have made corrections to the caption of Figure 4.]
Comments5:[In figure 7-8, the name of these genes should be italicized. Please check this error in this manuscript.]
Response5:[Thank you for your valuable suggestion. We have scanned the entire text and changed all gene names and IDs to italics.]
Comments6:[In Figure 8, I don't think the author's illustration is very clear about the difference between control and drought 0T. In “Expression level analysis of ZmLecRLK”, "after 10 days of exposure to drought stress" was described. However, we have no find 10 days in Figure 8. Please check and amend it carefully.]
Response6:[Based on your question, we have annotated all the symbols marked in the caption of Figure 8 to make it easier to understand.]
Comments7:[ Please check the format of references and amend it carefully.]
Response7:[We have checked the format of all references according to your request and made modifications to the incorrect format.]

Reviewer 3 Report
Comments and Suggestions for Authors
After a comprehensive review of the manuscript titled "Genome-Wide Analysis of Maize Lectin Receptor-Like Kinase (LecRLK) Family," I provide the following suggestions. The study represents a significant contribution to our understanding of plant molecular genetics, specifically addressing the structural and functional characterization of LecRLKs in maize. The research provides valuable insights into the genomic organization, evolutionary patterns, and potential functional roles of these important receptor-like kinases in plant cellular signalling and development.
Here are a few comments and suggestions-
1. Refine the title to be more specific and concise. The current title seems cut off and lacks clarity. Consider "Genome-Wide Characterization of Lectin Receptor-Like Kinase (LecRLK) Family in Maize: Structural, Evolutionary, and Functional Insights. "Refine the title to be more specific and concise. The current title seems cut off and lacks clarity. 2. Abstract Improvement Suggestions:
Clarify the specific scope of the study more precisely.Emphasize the novel contributions of the research. Provide more specific details about the methodology.
Recommendations: Briefly explain why LecRLK genes are essential in plant biology. Highlight the most significant findings more explicitly. Include a more explicit statement about the potential practical implications.
Language and Clarity: Reduce redundancy (e.g., "high and stable" is repeated). Use more precise scientific language. Break down complex sentences for better readability.
3. Structural and Content Suggestions: The introduction provides a comprehensive overview of LecRLK genes but could benefit from a more focused narrative flow. Consider reorganizing the paragraphs to create a more logical progression of ideas. Strengthen the rationale for why this study on maize LecRLK genes is novel and important 4. Emphasize the broader significance of LecRLK genes in crop improvement. Elaborate on how understanding these genes could contribute to maize breeding strategies. Provide more context about the unique characteristics of maize that make LecRLK research necessary. 5. Consider adding more recent citations to demonstrate the cutting-edge nature of the research 6. Highlight more explicitly the knowledge gap this research fills (currently mentioned briefly in lines 104-110) 7. Include more specific details about the potential implications of understanding LecRLK genes in maize.Lines 114-120- Specify the exact number of plants used in the study, Provide more details about the potting conditions and soil type, and Clarify the specific stress treatments and their durations
Recommendation: Add more methodological precision toplant cultivation
8. Lines 121-137: Clearly explain why Arabidopsis was chosen as a reference organism, Provide more detail about the criteria for sequence selection, and Clarify the rationale behind the 90% homology threshold.Recommendation: Include a flowchart or detailed description of the sequence identification process
9. Lines 184-195: Provide more details about selectingRNA-seq datasets, Explain the criteria for data quality control, and Clarify the significance of using different stress and tissue datasets. 10. Lines 174-182: Explain the significance of using a 2,000 bp upstream region, Provide more context about the importance of cis-regulatory elements, Clarify the criteria for miRNA target prediction 11. Lines 164-172: Explain the importance of comparing with Arabidopsis, rice, and sorghum; provide more context about why these specific organisms were chosen and clarify the significance of collinearity analysis in gene family studies.Recommendation: Add a brief explanation of how collinearity relates to gene evolution.
12. Lines 149-162: The choice of the Neighbor-Joining method for phylogenetic tree construction, Explain the significance of the 1000 bootstrap parameter, and Clarify the criteria for motif structure analysis.Recommendation: Include more details about the statistical robustness of the phylogenetic analysis
Author Response
Comments1:[Refine the title to be more specific and concise. The current title seems cut off and lacks clarity. Consider "Genome-Wide Characterization of Lectin Receptor-Like Kinase (LecRLK) Family in Maize: Structural, Evolutionary, and Functional Insights. "Refine the title to be more specific and concise. The current title seems cut off and lacks clarity. ]
Response1:[Thank you for your suggestion. We have changed the title of the article according to your suggestion.]
Comments2:[Abstract Improvement Suggestions:Clarify the specific scope of the study more precisely.Emphasize the novel contributions of the research. Provide more specific details about the methodology.Recommendations: Briefly explain why LecRLK genes are essential in plant biology. Highlight the most significant findings more explicitly. Include a more explicit statement about the potential practical implications.Language and Clarity: Reduce redundancy (e.g., "high and stable" is repeated). Use more precise scientific language. Break down complex sentences for better readability.]
Response2:[We have rewritten the abstract according to your suggestion.]
Comments3:[ Structural and Content Suggestions: The introduction provides a comprehensive overview of LecRLK genes but could benefit from a more focused narrative flow. Consider reorganizing the paragraphs to create a more logical progression of ideas. Strengthen the rationale for why this study on maize LecRLK genes is novel and important. ]
Response3:[We have revised the introduction according to your suggestion.]
Comments4:[ Emphasize the broader significance of LecRLK genes in crop improvement. Elaborate on how understanding these genes could contribute to maize breeding strategies. Provide more context about the unique characteristics of maize that make LecRLK research necessary. ]
Response4:[We have rewritten the last paragraph of the introduction according to your suggestion.]
Comments5:[ Consider adding more recent citations to demonstrate the cutting-edge nature of the research]
Response5:[We have added recent cutting-edge references as per your request.]
Comments6:[ Highlight more explicitly the knowledge gap this research fills (currently mentioned briefly in lines 104-110) ]
Response6:[We have made revisions to the article according to your suggestions.]
Comments7:[Include more specific details about the potential implications of understanding LecRLK genes in maize.Lines 114-120- Specify the exact number of plants used in the study, Provide more details about the potting conditions and soil type, and Clarify the specific stress treatments and their durations Recommendation: Add more methodological precision toplant cultivation.]
Response7:[We have added potting conditions and soil types in Materials and Methods 2.1 as per your suggestion. The reference literature on coercion treatment and duration has been provided in section 2.1.]
Comments8:[Lines 121-137: Clearly explain why Arabidopsis was chosen as a reference organism, Provide more detail about the criteria for sequence selection, and Clarify the rationale behind the 90% homology threshold.Recommendation: Include a flowchart or detailed description of the sequence identification process]
Response8:[We have made modifications to Materials and Methods 2.2 as per your suggestion.]
Comments9:[Lines 184-195: Provide more details about selectingRNA-seq datasets, Explain the criteria for data quality control, and Clarify the significance of using different stress and tissue datasets. ]
Response9:[We have made modifications to Materials and Methods 2.7 as per your suggestion.]
Comments10:[Lines 174-182: Explain the significance of using a 2,000 bp upstream region, Provide more context about the importance of cis-regulatory elements, Clarify the criteria for miRNA target prediction]
Response10:[We have made modifications to Materials and Methods 2.6 as per your suggestion.]
Comments11:[ Lines 164-172: Explain the importance of comparing with Arabidopsis, rice, and sorghum; provide more context about why these specific organisms were chosen and clarify the significance of collinearity analysis in gene family studies.Recommendation: Add a brief explanation of how collinearity relates to gene evolution.]
Response11:[We have made modifications to Materials and Methods 2.5 as per your suggestion.]
Comments12:[Lines 149-162: The choice of the Neighbor-Joining method for phylogenetic tree construction, Explain the significance of the 1000 bootstrap parameter, and Clarify the criteria for motif structure analysis. ]
Response12:[We have made modifications to Materials and Methods 2.4 as per your suggestion.]

Reviewer 4 Report
Comments and Suggestions for Authors
The study of kinase receptors is an important and interesting task, since kinase receptors are involved in the transmission of various signals, especially in response to any impacts on plants. It is especially important to identify which receptors are involved in response to which stressors. The presented manuscript is devoted to these important problems. The manuscript is carefully designed, a large experimental material. There are some minor comments.
In the materials and methods section, it is necessary to indicate at what stage of maize development the material was collected. It is also necessary to note which organs and tissue types were collected for analysis.
Figure 3 - what is the scale on the left side of the figure?
495 - repetition of a sentence
501 - where are the tissue types indicated?
Author Response
Comments1:[ In the materials and methods section, it is necessary to indicate at what stage of maize development the material was collected. It is also necessary to note which organs and tissue types were collected for analysis.]
Response1:[The above content can be found in Materials and Methods 2.1. The collection and development stages of corn materials, as well as the types of organs and tissues collected, refer to the work of Cristian and Swat.]
Comments2:[ Figure 3 - what is the scale on the left side of the figure?]
Response2:[Your question can be found in the caption of Figure 3.]
Comments3:[495 - repetition of a sentence]
Response3:[I'm very sorry, we couldn't find a duplicate sentence on line 495. If you are certain that there are duplicate sentences, please highlight them and we would greatly appreciate it.]
Comments4:[501 - where are the tissue types indicated?]
Response4:[Your question can be answered in Figure 7.]

Round 2
Reviewer 2 Report
Comments and Suggestions for Authors
Thank you for your careful revision.
Reviewer 3 Report
Comments and Suggestions for Authors
The authors have carefully consider my comments and addressed the concerned. I have no additional comments and recommend publication of this article.
Congratulations!